# Research on Geometric Constraint Strategies for Controlling the Diameter of Micro-Shafts Manufactured via Wire Electric Discharge Grinding

**DOI:** 10.3390/mi14122178

**Published:** 2023-11-29

**Authors:** Jianyu Jia, Zan Li, Bo Hu, Yanqing Wang, Jing Wang, Congbo Li, Wenfeng Xiang

**Affiliations:** 1Key Laboratory of Precision Machining of Shanxi Province, College of Mechanical and Vehicle Engineering, Taiyuan University of Technology, Taiyuan 030024, China; jiajianyu@tyut.edu.cn (J.J.); lizan_k@163.com (Z.L.); hubo0156@link.tyut.edu.cn (B.H.); wangjing02@tyut.edu.cn (J.W.); 2State Key Laboratory of Mechanical Transmission, College of Mechanical and Vehicle Engineering, Chongqing University, Chongqing 400044, China; 3Taizhou Jiuju Technology Co., Ltd., Taizhou 317600, China; xiangwenfeng2005@163.com

**Keywords:** micro EDM, WEDG, micro-shaft, geometric constraint, twin-mirroring-wire electrodes, diameter control

## Abstract

Micro-tools comprising difficult-to-machine materials have seen widespread application in micro-manufacturing to satisfy the demands of micro-part processing and micro-device development. Taking micro-shafts as an example, the related developmental technology, based on wire electric discharge grinding (WEDG) as the core method, is one of the key technologies used to prepare high-precision micro-shafts. To enable efficient and high-precision machining of micro-shafts with target diameters, instead of performing multiple repeated on-machine measurements and reprocessing, a geometric constraint strategy is proposed based on the previously introduced twin-mirroring-wire tangential feed electrical discharge grinding (TMTF-WEDG). This strategy encompasses the tool setting method, tangential feed distance compensation, and an equation that establishes the relationship between tangential distance and diameter variation. These components are derived from a key points analysis of the geometric constraints. The micro-shafts with diameters of 50 µm and consistencies of ±1.5 µm are repeatedly processed. A series of micro-shafts with diameters ranging from 30 µm to 120 µm achieve geometric constraints with a diameter accuracy of ±2 µm, accompanied by the complete continuous automation of the entire process. Accordingly, it can be concluded that the geometric constraint strategy is flexible and stable and can be controlled with high precision in the TMTF-WEDG process.

## 1. Introduction

The demand for micro-parts and micro-devices with micro features has increased across multiple industries, finding prominence in the fields of biotechnology, medical services, microelectronics, and precision instruments [1,2,3,4]. Micro-features with micron-characteristic sizes and sub-micron accuracy are primarily processed using micro-machining technologies. This requires the application of micro-tools. For example, micro-cutting/drilling/milling/grinding tools are required to perform mechanical micro-machining. Conversely, micro-electrodes are necessary for micro-electrochemical machining (micro-ECM) and micro-electrical discharge machining (micro-EDM) [5]. The materials used for micro-tools are usually difficult-to-machine materials, such as cemented carbide, polycrystalline diamonds (PCD), and cubic boron nitride (CBN). Micro-shafts and similar parts, being the most common micro-tools, are widely used in micro-operations and micro-machining [6,7,8], such as the processing of micro-holes, micro-grooves, and micro-three-dimensional structures on the surface. The accuracy of the structures or parts to be processed depends on the uniformity of the micro-shafts, including the uniformity of the diameter in the axial direction of a single micro-shaft and the repeatability of the diameter for repeatedly processed micro-shafts. Micro-shafts can be directly applied and processed into different special structures or subsequently processed to meet different application requirements.

There are several techniques available for the fabrication of micro-shafts and similar parts. Micro-EDM has emerged as a leading technique because of its flexibility, economic viability, non-contact nature, and capacity to machine materials with high hardness, strength, and temperature resistance [8,9,10,11]. In particular, the wire electrical discharge grinding (WEDG) method proposed by Masuzawa et al. [12] in 1985 makes micro-EDM more suitable for use in the high-precision machining of micro-shafts because of the lower energy transmitted via point discharge and wire electrode refreshment. Over the years, extensive new theoretical concepts, technical methods, and processing equipment for WEDG-based processes of micro-tool fabrication have been proposed and developed successively.

The micro-shafts processed by Lim et al. [13] using the WEDG method did not efficiently achieve adequate surface-processing quality, and it has been pointed out that the speed of the finishing process must be slow. With the assistance of an inchworm type of micro-feed mechanism, Li et al. [14] successfully fabricated a micro-shaft with a diameter of 25 µm and an aspect ratio of 20. Based on a three-axial CNC EDM machine, Huang et al. [15] developed WEDG machining equipment and processed micro-shafts that were 2 mm long and either 30 µm or 50 µm in diameter. Wu et al. [16], Chern et al. [17], and Kuo et al. [18] continued to manufacture high-aspect-ratio cut-side micro-shafts, noncircular cross-sectional micro-shafts, and series-pattern micro-disk electrodes based on the micro-shafts processed via WEDG, respectively. Rees et al. [19], Sun et al. [20], and Li et al. [21] applied special strategies to achieve efficient and high-quality micro-shafts machining, respectively, such as combining WEDG with micro-wire electrical discharge machining (micro-WEDM), performing low-speed wire electrical discharge turning (LS-WEDT) combined with the multiple cutting strategy, and enabling the active supply of wire electrodes in WEDG (AS-WEDG).

Rees et al. [22] also reported that higher-quality surface finish and dimensional accuracy can be achieved by applying a dressing strategy that relies on multiple reciprocating feeds of the micro-shafts during the WEDG. Variations in the setting of the dressing position occur owing to the positioning accuracy and the repeatability of the EDM machine, especially when the electrodes are brought into contact with the micro-shaft. Wang et al. [23] discovered that the guided running wire electrodes can cause fluctuations in the front edge of the machining area during the WEDG, which may be the cause of the diameter deviation of micro-shafts. They proposed the strategy of zero infeed and stopping the running of the wire electrode, which were applied to improve the fabrication repeatability of micro-shafts in the range of ±2 μm via WEDG. An improved WEDG method was developed by Zou et al. [24]. It processes a new feature of a positioning device to address the wire vibration problem, enabling the method to enhance the micro-shafts fabrication precision of 14 µm diameter micro-shafts with less than 0.4 µm deviation and an aspect ratio of 142. Tangential feed WEDG (TF-WEDG) can achieve micro-shafts of high accuracy within 1 μm and repeatability within 2 μm, as proposed by Zhang et al. [25], by reducing the effect of the positioning error compared to the conventional radial-feed WEDG process.

However, the on-process measurement using optical measuring devices is essential to controlling the diameter accuracy of micro-shafts for continuous processing and subsequent applications, which limits the ability of WEDG to accurately shape micro-shafts. Sheu [26] developed a hybrid-circuits twin-wire WEDG (HCTW-WEDG) system. It can reduce the processing time of micro-shaft fabrication by two-thirds via simultaneously conducting roughing and finishing. Meanwhile, the control of the diameter accuracy of the micro-shaft can only be achieved via controlling the gap distance of the twin wire. The new method presented by Wang and Jia et al. [27,28], named twin-mirroring-wire tangential feed electrical discharge grinding (TMTF-WEDG), is mainly characterized by the tangential feed strategy and the narrow slit formed using a twin-mirroring wire to shape the micro-shaft using geometric constraints.

In order to avoid multiple repeated online measurements and reprocessing and to facilitate the efficient and high-precision machining of target-diameter micro-shafts arbitrarily, further research has been carried out based on the TMTF-WEDG method. The specific aim was to achieve geometric constraints on the diameter of the micro-shafts and solve several specific problems that affect diameter prediction and control in this process.

## 2. The TMTF–WEDG Method and Analysis

### 2.1. The TMTF–WEDG Method and Device

The TMTF–WEDG method and device are presented in Figure 1 and Figure 2, respectively, as mentioned in our previous research [27,28], and the diameter of the micro-shafts is geometrically constrained by the axial/tangential feed motion of micro-shafts and the narrow gap formed using twin–mirroring–wire electrodes. The step of measuring the diameter of the micro-shafts mid–process and determining the machining allowance to continue machining is unnecessary and can be omitted.

In the XY plane, two wire electrodes are constrained at the same horizontal height and insulated from each other. The width of the formed narrow slit gradually decreases with the growing distance of tangential feed (in the Y direction). Therefore, the key to the constrained formation of the micro-shafts lies in its tangential feed distance along the symmetrical center line of the narrow slit. The reciprocating motion along the z± of the machine tool determines the length of the micro-shafts. The entire manufacturing process is divided into three stages: roughing, semi–finishing, and finishing. The important parameters of the TMTF–WEDG machining process are listed in Table 1. The SEM photos of the processed micro-shaft and the surface topography of each stage are shown in Figure 3. The surface roughness of the micro-shaft is below Ra 0.12 μm, and the diameter of the electric melting pits ranges from only 2 to 4 μm.

The division of the three stages of TMTF-WEDG is based on the matching relationship between the diameter deviation range and the machining allowance of each stage. In Figure 4, it can be seen that during the roughing stage (with tangential feed steps of 1–10), the maximum upper and lower deviations of the diameter of the micro-shaft are 21.7 μm and 22.3 μm, respectively, while these two values decrease to 11.07 μm and 9.88 μm in the semi–finishing stage (with steps 11–13). Finishing (steps 14–16) gradually improves the diameter consistency of the micro-shafts, with upper/lower deviations of 5.43 μm/5.96 μm, 2.29 μm/2.88 μm, and 1.32 μm/1.47 μm, respectively. Therefore, the allowances on a radius for semi-finishing and finishing should be kept separate by at least 22 μm and 11 μm. More meaningfully, the result serves as a reference for determining the tangential feed distance or final machining position of each stage. It should be noted that since the experimental data are a preliminary exploration, the diameter accuracy control strategies of the micro-shafts have not yet been adopted. The purpose of experimental data is to clarify a reference for determining the tangential feed distance and to test whether the selected processing parameters are applicable.

### 2.2. Keypoints Analysis of Geometric Constraints on the Diameter of Micro-Shafts

From the TMTF-WEDG machining method, it can be perceived that the key points of geometric constraints on the diameter of micro-shafts contain four aspects:The tangential feed path of the micro-shafts should be the symmetrical centerline of the narrow slit formed via the twin–mirroring–wire electrodes;The diameter of the micro-shafts is controlled by the distance of the tangential feed;The width of the narrow slit can be precisely controlled and adjusted to achieve different minimum limit values of micro-shaft diameters;The boundary on both sides of the narrow slit, namely the boundaries of the twin–mirroring–wire electrodes, should be stable and free from fluctuations.

For the first key point, the symmetrical centerline of the twin–mirroring–wire electrodes can be measured via on–machine CCD. Figure 5 presents the CCD photos of the narrow slit formed via twin–mirroring–wire electrodes. The position curves of the narrow-slit contour and the symmetrical centerline relative to the machine tool can be obtained via the pixel binarization of the captured images of the boundaries of the twin-mirroring-wire electrodes and processing them using MATLAB, as reported in our previous research [28]. This means that the tangential feed path of the micro-shafts can also be accurately captured despite installation errors in the device.

Meanwhile, the width of the narrow slit can be accurately determined as an equation of the tangential feed distance. The width curve A in Figure 6 describes the width variation in the narrow slit along the tangential direction (Y–axis) in Figure 5, and the width curve of the narrow slit can be fitted using polynomials:*y* = −6E − 10*x*^3^ + 2E − 05*x*^2^ − 0.1533*x* + 295.97(1)
where *y* represents the width of the narrow slit, and *x* represents the tangential position coordinates.

The correlation coefficient *R*^2^ of the fit degree can reach 0.9998.

Significantly, the result of the tool setting determines whether the rotation center of the micro-shafts can be located on the path of the symmetrical centerline. Additionally, this factor affects the distance of the tangential feed. A detailed discussion on tool setting might be conducted in Section 3.1.

In Figure 6, the initial width of the narrow slit is 18 μm (width curve A), and the width of the adjusted narrow slit is 40 μm (width curve B). The adjustment of the narrow-slit width is achieved by pushing the flexible hinge with a preloaded piezoelectric ceramic actuator (P–844.60 made using PI: travel 90 μm; propulsive force 3000 N; and min. resolution 0.9 nm) equipped with the constrained guide wheel A. Adjusting the narrow-slit width has little effect on the tangential feed path, as the path is only translated in the X–axis direction at half the adjustment amount, as shown in Figure 7a.

However, the initial tool-setting position, the tangential feed distance, and the minimum diameter of micro-shafts to be machined may undergo significant changes, as illustrated in Figure 7b. Under the condition of adjusting the width of narrow slits at the micrometer level, the entire machining range of micro-shafts is always within the envelope angle range of the wire electrodes on the constraint guide wheel. Taking the processing of micro-shafts with a diameter of 45 μm as an example, Table 2 lists the data based on Figure 6 and obtained via two different narrow-slit widths, A and B.

On the one hand, the shorter the tangential feed distance, the shorter the processing time. Conversely, adjusting narrower slits is beneficial for machining micro-shafts with different target diameters and smaller diameters by controlling the tangential feed distance without the need to frequently adjust the width of the slits.

Additionally, there is an obvious geometric relationship in Figure 7:(2)$${(S_{1}/2+R_{W})}^{2}+{\left(F-f\right)}^{2}={\left(r+R_{W}\right)}^{2}$$
(3)$$r=\sqrt{{\left(S_{1}/2+R_{W}\right)}^{2}+{\left(F-f\right)}^{2}}-R_{W}$$

The last key point, the stability of the narrow-slit boundary, is directly related to the running state of the wire electrodes. Wire electrode A and wire electrode B are driven by the same wire-winding motor to unify the running velocity. Different wire electrodes running linear velocities can be achieved by setting the control variables of the wire-receiving servo motor. Figure 8 depicts the fluctuations in wire electrode A and wire electrode B at different linear velocities, and detailed average values are listed in Table 3.

From the above data, it is possible to prove that the sum of the fluctuations in the twin-mirroring-wire electrode is the smallest, which means that the boundary of the narrow slit is the most stable when the linear velocity is 107 μm/s. To ensure that the diameter accuracy of the micro-shafts is not affected by the fluctuation in the wire electrodes, the twin-mirroring-wire electrodes remain stationary while maintaining tension during finishing. Nevertheless, static wire electrodes cannot achieve the compensation of discharge wear. The relationship between the diameter change of the micro-shafts and the wear of the wire electrodes is the focus of the subsequent content (Section 3.2).

In summary, geometric constraints of the diameter of the micro-shafts processed via WEDG can be achieved using the twin-mirroring-wire electrode by clarifying the tangential feed path, the variation law of the narrow-slit width, and the stability of the narrow-slit boundary. Moreover, the details of the geometric constraint strategy need to be further discussed.

## 3. Geometric Constraint Strategy for Micro-Shaft Diameter Control

To achieve geometric constraints via the application of TMTF-WEDG to the diameter of micro-shafts, specific strategies need to be proposed, including the tool-setting method for the micro-shaft blank, the compensation for tangential feed distance, and the relationship between wire-electrode wear and the variation in the micro-shaft diameter.

### 3.1. Tool-Setting and Tangential Feed Distance

The purpose of the tool setting for TMTF-WEDG is to clarify the initial machining position and ensure that it is located on the symmetrical centerline. The process of tool setting is presented in Figure 9. The waveform characteristics of the contact between the micro-shaft blanks and the wire electrodes A and B can be used as the standard for tool setting. Electrical signals of the contact between the micro-shaft blank and the wire electrodes are collected and uploaded via the contact-sensing circuit. When the difference between the average contact voltage *u*_a_ between the blank and the wire electrode A and the *u*_b_ between the blank and the wire electrode B is within 0.5 V, it can be considered that the rotation center of the blank is located on the symmetrical centerline.

Significantly, the blank is clamped onto the rotating spindle (rotational runout less than 1 μm), so several errors may occur. Figure 10a indicates the factors affecting the effective diameter of the blank, which determines the tool-setting position, including the clamping error (*Fe*) between the geometric axis of the micro-rods and the rotation axis of the spindle, the diameter tolerance of the rod blank (*Be*), the radial runout of the spindle (*Se*), and the title angle of the rods (*θ*). Under the effects of these four factors, the effective diameter *De* of the blank is:*De* = 2 (*Fe* + *Be*/2 + *Se* + *l* tan *θ*) +*Dn*(4)
where *l* and *Dn* are the length and the nominal diameter of the blank, respectively.

Obviously, under the effects of multiple error factors, the blank makes contact with the wire electrode A and the wire electrode B periodically, as shown in Figure 10b. After the contact with the wire electrode A, the blank rotates through the *T_NC1_* arc segment. Similarly, the rod blank rotates through the *T_NC2_* arc segment during contact with the wire electrode B and then repeats the above process periodically. Due to the narrow slit formed by the wire electrode appearing in a trumpet shape in the XY plane, the arc length corresponding to *T_NC1_* is greater than that corresponding to *T_NC2_*. If the blank is exactly on the symmetrical line of the twin-wire electrodes, the length of the *T_C1_* is equal to the length of the *T*_C2_, as depicted in Figure 10c. Additionally, the length of the *T_C1_* and the *T*_C2_ depends on the overfeed along the Y + direction. The greater the overfeed, the greater the contact length of the arc segment.

Figure 11a presents the tool-setting waveforms between the blank and the twin wires. When the output voltage is at a high level, the workpiece and the wire electrode are in contact. When it is low, the two are in a non-contact state. There is a phase difference between the two contact waveforms. It is necessary to adjust the position of the blank in the X direction so that the contact time *T_C1_* = *T_C2_*, indicating that the rotation center of the micro-rod lies on the symmetrical line of the slit. It can be seen from Figure 11 that the time T_NC1_ > T_NC2_ is consistent with the theoretical analysis of the tool-setting waveform. In addition, there is a relationship between the period *T_A_* and *T_B_*: *T_A_* (*T_C1_*+*T_NC1_*) = *T_B_* (*T_C__2_*+*T_NC__2_*) ≈ 50 ms, corresponding to a spindle speed of 1200 rpm.

It can be seen from Figure 11b that the duty cycle of the tool-setting waveform ranges from 23% to 70% and that the duty cycle stabilized at around 30% within the first 10 s. The twin-wire electrodes were kept in a static state during the tool-setting process to avoid the effect of wire fluctuation. The sliding and wear between the blank and the wire electrode cause this phenomenon, which can be ignored because the actual tool-setting process is completed in just several seconds. Thereby, the initial machining position of the micro-shafts is determined, which is crucial for determining the tangential feed distance of the micro-shafts and achieving geometric constraints on the diameter.

Taking the 30% duty cycle as an indicator, we performed an assessment. We found that an inconsistent error between different blanks results in a floating initial machining position after the tool setting, as described in Figure 12a. In Figure 12b, assuming that position *I* corresponds to the rotation center point with a narrow-slit width of 500 μm, *I_1_* may be described as the actual initial machining position. Therefore, the tangential feed distance will also change from *f* to *Σf*, which indicates that error factors increase the diameter of the rotating circle. With the increase in the effective diameter, the tool-setting position becomes far away from the position where the narrow-slit width is the smallest.

It can be proven from Figure 13a that there exists a deviation *v* between the geometric center and the rotation center of the blank. By conducting a statistical analysis on 15 randomly selected blanks in Figure 13b, the rotation diameter of blanks is mainly affected by the deviation *v.* The test results of 15 blanks show that the diameter tolerance is about ±5 μm and that the clamping error may go up to 20 μm. The maximum difference between the rotation and geometric diameters can reach 46 μm.

Changes in tangential feed distance Δ*f* can be further investigated by allowing 5 out of 15 blanks to perform a discharge of short duration via the wire electrode at the tool-setting position. This engraves the initial machining position. Figure 14a,b record the tool-setting position and the contour curves of the wire electrodes, respectively. Figure 14b shows the relative position of the center of the rotation circles of the different blanks. The relative position of the tool-setting position in the CCD photos is recorded in Figure 14c. The variation in the tool-setting positions in the tangential direction is close to 60 μm.

If the problem of initial machining position floating and tangential feed distance compensation is not solved, on the one hand, the actual machining diameter of the micro-shafts is greater than the target value. Conversely, the consistency of the accuracy of the diameter of the repeated machined micro-shafts cannot be guaranteed in such scenarios. After unifying the relative positions in Figure 14 with the coordinate system in Figure 6, Figure 15a reveals the coordinates of the five initial machining positions on the tangential feed path. If the position with a narrow-slit width of 500 μm is set as the ideal initial point, the tangential feed distance of the five blanks tested in the experiment must be compensated, and the compensation values are different, as shown in Figure 15b.

In conclusion, a tool setting and compensation method for tangential feed distance has been proposed. The problem of tool position fluctuation caused by errors during tool setting and the change in tangential feed distance resulting from initial machining position fluctuation has been effectively solved.

### 3.2. The Relationship between the Material Removal Amount of Micro-Shafts and Wire-Electrode Wear

During the finishing stage of TMTF-WEDG, the wire electrodes remain stationary under the tension force. With the tangential feeding of micro-shafts, the wire electrodes are inevitably discharged and become worn, as described in Figure 16. Accordingly, if the geometric constraints are imposed on the micro-shaft diameter by the narrow slit formed via twin-mirroring-wire electrodes, it is necessary to clarify the relationship between the material removal amount (MRA) of micro-shafts and wire-electrode wear (WEW).

Due to the characteristics of a gradually decreasing narrow slit and the effect of tangential feed on improving a material removal resolution, the MRA and WEW also exhibit a gradual change accordingly. Then, the experiments of finishing micro-shafts via a single-axial feed are conducted at different positions on the tangential feed path. Furthermore, the continuous tangential feed micro-shaft model is implemented in finishing to reveal the quantitative relationship between MRA and WEW.

Calculating the material removal amount for micro-shafts is relatively simple, based on the volume difference before and after processing. The calculation of relative wire-electrode wear is shown in Figure 17. By performing CCD photography and Matlab image processing to obtain the boundary of wire electrodes before and after processing, the diameter change of the wire electrodes at each pixel point can be calculated as altering from *λ* to *ζ*. The product of the wear area (black shaded area) and pixel width of each wire electrode slice is the volume of that slice. The total wear amount is the sum of the wear volume of each slice. Although the angles corresponding to different positions change with the tangential feed of the micro-shafts, such as changing from *θ*_0_ at the initial position to *θ_f_* at the end position, the change in angle exerts almost no effect on the calculation of wire-electrode wear. This is illustrated by the fact that the curvature radius of the wire electrode is 9.2 mm and the pixel width is only 0.345 μm, meaning that the (*θ_Δ2_−θ_Δ1_)* is 7.94″.

Figure 18 and Figure 19 present the micro-shaft results with diameters of 80 μm and 60 μm, respectively. It can be observed that as the tangential feed distance increases, the diameter change of the micro-shaft becomes smaller and smaller after a single axial feed, indicating a decrease in the material removal amount (reducing from 439,268.26 μm^3^ to 180,188.22 μm^3^) and an improvement in the material removal resolution.

For wire-electrode wear, the contour can be reasonably fitted using circular curves. This requires the application of 1stOpt (15PRO) software, and the results are shown in Figure 18e and Figure 19e. The fitting procedure is as follows:

Variable X,Y,U,V;

Parameters a,b,r;

Function Y = −SQRT(r^2 − (X + a)^2) − b;

Function V = SQRT(r^2 − (U + a)^2) − b;

Data;

//X,Y,U,V

0,258.405,0,322.92

…

The fitting results of the wire-electrode wear in Figure 18 and Figure 19 can be expressed using the following equations:(*x* − 26.744)² + (*y* − 365.029)²= 38.868²(5)
(*x* − 27.501)² + (*y* − 359.186)² = 30.732²(6)

In Figure 18, the arc diameter in the wear area of the wire electrode is 77.736 μm, whereas the average diameter of the micro-shaft after processing is 76.218 μm. The difference between the diameter of the fitted circle and the average diameter of the micro-shaft after machining is 1.518 μm. The three values in Figure 19, respectively, are 61.464 μm, 58.375 μm, and 3.089 μm. However, the width variation in the narrow slit in both positions reaches 10 μm, suggesting that the wire-electrode wear amount could be similar. According to the calculation method of the wire-electrode wear amount shown in Figure 17, the wire-electrode wear in Figure 18 is 12,156.02 μm^3^, whereas it is 11,133.66 μm^3^ in Figure 19. Therefore, it can be concluded that the ratio of MRA to WEW is not fixed but rather varies with the tangential position. Obviously, the larger the *f* is, the smaller the ratio will be. To further clarify the regular pattern of micro-shaft diameter variation during finishing, it is necessary to study MRA and WEW during the continuous tangential feed process.

Figure 20a shows the CCD photo of the worn twin-mirroring-wire electrode with a tension force and a stationary state during the finishing of TMTF-WEDG, in which the wear amounts of both wire electrodes appear similar. The total finishing tangential feed distance is approximately 609 μm, corresponding to a change in a center angle (*θ*_0_ − *θ_f_*) of 3.805°. The variation in narrow-slit width before and after precision machining and the variation in the micro-shaft diameter with tangential feed is described in Figure 20b. The micro-shaft diameter change curve is practically consistent with the narrow-slit width curve of the worn-wire electrode. On the contrary, the diameter variation in the constrained micro-shafts does not follow the variation pattern of the narrow-slit width.

Figure 21 presents the wear of wire electrodes A and B and the micro-shaft diameter change, following the method shown in Figure 17. The average diameter of the micro-shaft is processed from 98.76 μm to 34.38 μm. The total MRA is 6,498,401.898 μm^3^, while the WEW is 908,382.85 μm^3^, and their ratio is 7.154.

The geometric constraint of the diameter of micro-shafts is not consistent with the relationship between the width of the narrow slit and the distance of the tangential feed. This can be attributed to the fact that wire electrodes remain stationary under tension to ensure diameter accuracy in the TMTF-WEDG finishing process. By studying the relationship between the material removal amount of micro-shafts and the wear amount of the wire electrodes, the variation regulation of the micro-shaft diameters with the wear state of the wire electrodes is obtained. This means that even if the wear of the wire electrode exists, the geometric constraint of micro-shafts can be completed.

In Figure 22, the diameter variation process of ten repetitively machined micro-shafts with a tangential feed shows a certain relationship with the variation in the narrow-slit width. In Figure 23, the actual and theoretical values correspond to the average diameters of the 10 micro-shaft diameters in the different tangential positions and narrow-slit widths in Figure 22, respectively. There is no unexpected deviation between the actual and the theoretical values within the finishing range due to wire-electrode wear. Therefore, the geometric constraint equation of the actual diameter of the micro-shafts changes.

Based on theoretical Equation (3) and data in Figure 22, the geometric constraint equation between the tangential feed distance *f* and the actual target diameter *d* of the micro-shaft can be fitted using 1stOpt as:(7)$$d=2\left(\sqrt{7819.12^{2}+{\left(1988.58-f\right)}^{2}}-\;7808.3\right)$$

The fitting procedure is as follows:

Variable f,d;

Parameters a,b;

Function d = 2*(SQRT(a^2 + (1988.58 − f)^2) − b);

Data;

//f,d

−28.13, 507.9529075

91.014, 462.91341

211.57, 415.3710725

…

The fitted value curve is shown in Figure 23. The mean-square deviation between the fitted and actual values is 7.7186, and the correlation coefficient can reach 0.9987. The equation can support geometric constraints for the repetitive processing of micro-shafts with different diameters using TMTF-WEDG.

## 4. Repetitive Machining of Micro-Shafts with Different Diameters

Based on the aforementioned research results, repetitive machining experiments were conducted on the diameter of micro-shafts with geometric constraints using TMTF-WEDG, as shown in Figure 24 and Figure 25.

Figure 24a presents the SEM photos of four micro-shafts with a target diameter of 50 μm and actual diameters of 50 ± 1.5 μm. The diameter curves of the micro-shafts, obtained via CCD photography and Matlab image processing, are shown in Figure 24b. Taking the tangential position in the feed path with a narrow slit width of 500 μm as the origin, the theoretical tangential feed distance *f* is 1517.5 μm. Table 4 shows the statistics of Δ*f* and *Σf.* The result proved that using the geometric constraint strategy to accurately achieve micro-shaft diameter repeatability is possible, supported by tool-setting methods, tangential feed distance compensation, and constraint equation.

Figure 25 demonstrates the repetitive machined micro-shafts with different diameters, such as target diameters of 30 μm, 40 μm, 60 μm, 80 μm, 100 μm, and 120 μm, respectively. According to the data of micro-shafts with different diameters recorded in Figure 25b, the control of the target diameter was achieved within a tolerance range of ±2 μm. Among the tested objects, sample b is a kind of stepped micro-shaft with diameters of 40 μm and 80 μm. Table 5 provides the average diameter after processing, the tangential feed distance, and the compensation values of different micro-shafts during processing. It can be concluded that the TMTF-WEDG method supported via the geometric constraint strategy has the ability to control the micro-shaft diameter in relation to different target values with relatively high diameter consistency accuracy.

Eventually, the diameter of micro-shafts is geometrically constrained through the narrow slit formed by the stationary wire electrode under a tension force based on the TMTF-WEDG method. Due to the significant increase in material removal for manufacturing ultra-thin disc-cutting tools using TMTF-WEDG, the wear of the wire electrodes also significantly increases. Running wire electrodes will be necessary to compensate for wear, while the strategy of keeping the wire electrode stationary in finishing will no longer be applicable. The constant tension control system of wire electrodes will be effective for future research issues.

## 5. Conclusions

The proposed geometric constraint strategy solves several matters in the TMTF-WEDG process, arbitrarily achieving flexible and high-precision control of the diameter of micro-shafts. The main conclusions are as follows:In the TMTF-WEDG process, the narrow slit formed via twin-mirroring-wire electrodes can geometrically constrain the micro-shaft diameter arbitrarily by controlling tangential feed distance. The geometric constraint strategy includes the tool-setting method, the tangential feed distance compensation, and the equation for the relationship between tangential distance and diameter variation;The geometric constraint strategy solves the matter of the tangential feed distance change caused by the fluctuation in the tool-setting position and the wire wear caused by the wire electrodes being stationary under a tension force to maintain the high accuracy of micro-shafts;Supported via a geometric constraint strategy, both high-precision repetitive machining of micro-shafts with a certain diameter and any target diameter are flexibly achieved without any interruption in the process. For instance, the result shows four micro-shafts with diameters of 50 µm and a consistency of ±1.5 µm and a series of micro-shafts with diameters ranging from 30 µm to 120 µm and a diameter accuracy of ±2 µm.

## Figures and Tables

**Figure 1 micromachines-14-02178-f001:**
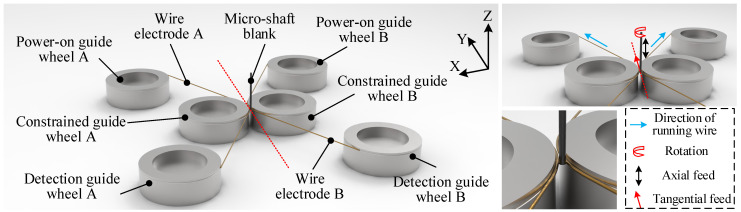
The method of TMTF-WEDG.

**Figure 2 micromachines-14-02178-f002:**
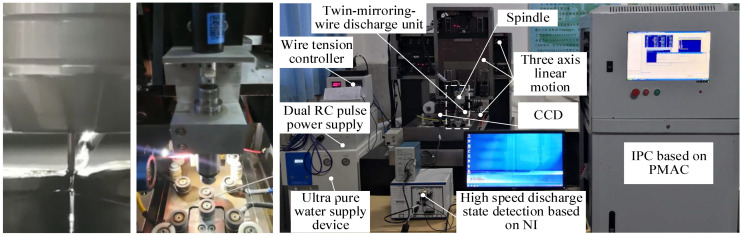
The device of TMTF-WEDG.

**Figure 3 micromachines-14-02178-f003:**
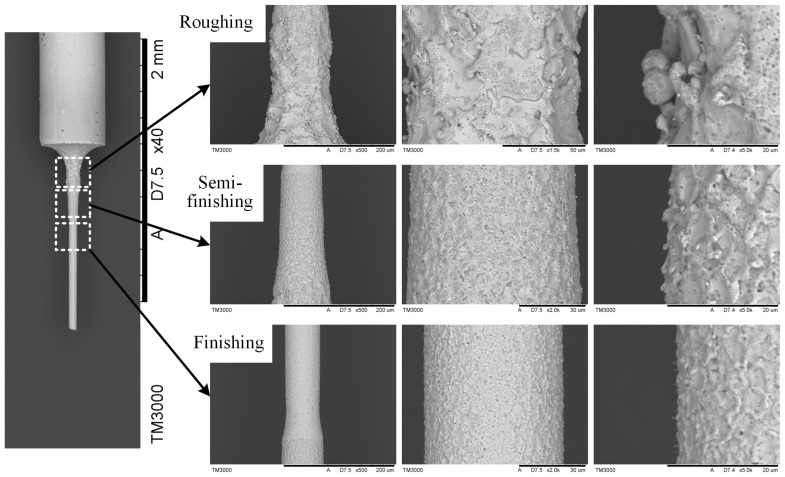
The SEM photos of the micro-shaft and the surface topography of each stage.

**Figure 4 micromachines-14-02178-f004:**
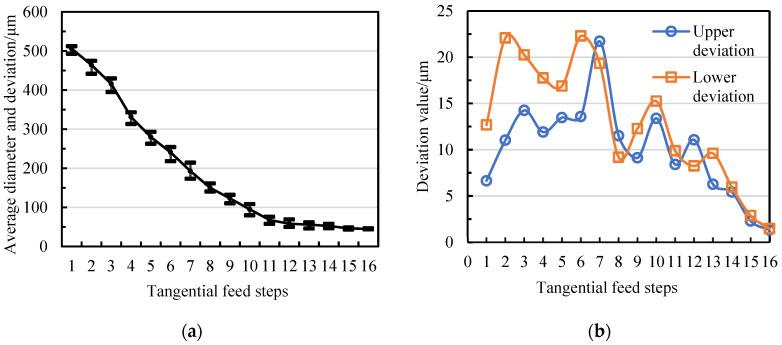
Diameter deviation of the micro-shaft at each machining position: (**a**) average diameter and deviation variation in micro-shafts and (**b**) upper and lower deviation of the micro-shaft diameter.

**Figure 5 micromachines-14-02178-f005:**
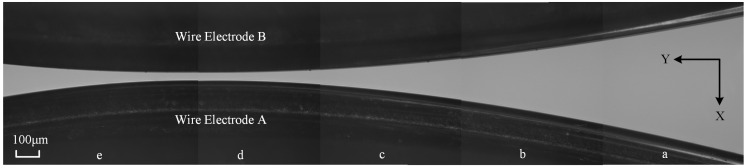
CCD photos of the narrow slit.

**Figure 6 micromachines-14-02178-f006:**
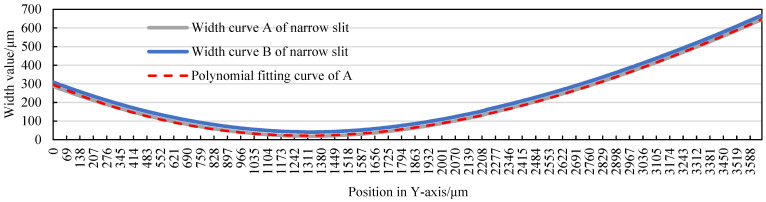
The curves of the width of the narrow slit and polynomial fitting.

**Figure 7 micromachines-14-02178-f007:**
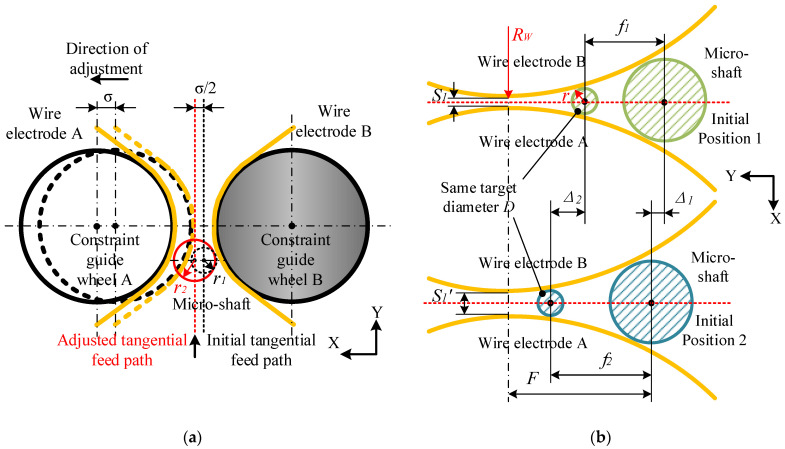
The influence of narrow-slit width adjustment on the TMTF-WEDG process: (**a**) on the tangential feed path and (**b**) on the initial position, the tangential feed distance, and the minimum diameter.

**Figure 8 micromachines-14-02178-f008:**
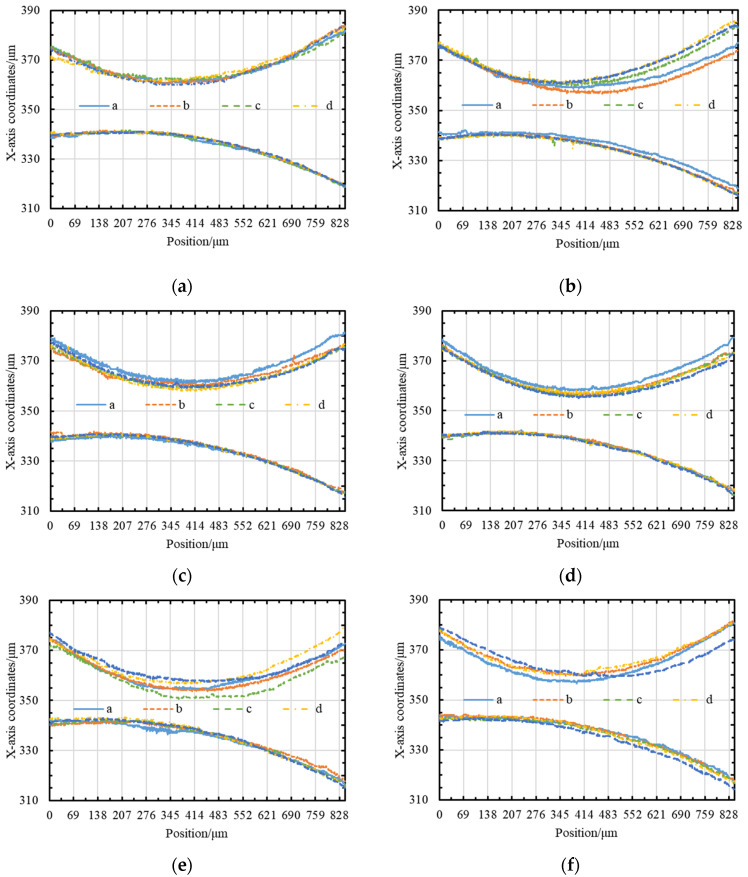
The fluctuation in twin-mirroring-wire boundaries at different linear speeds: (**a**) 107 μm/s; (**b**) 213.5 μm/s; (**c**) 427 μm/s; (**d**) 640.5 μm/s; (**e**) 854 μm/s; and (**f**) 1067.5 μm/s.

**Figure 9 micromachines-14-02178-f009:**
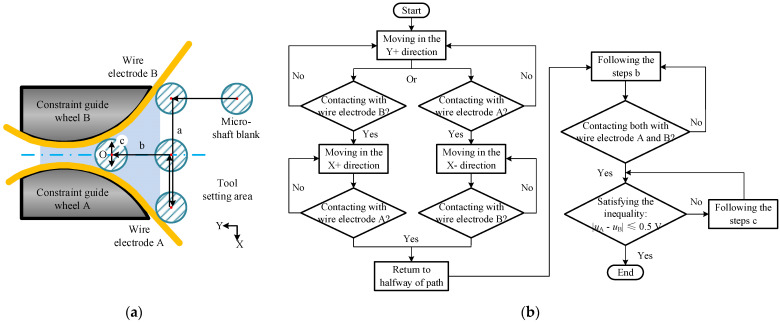
The method of tool setting: (**a**) schematic diagram of the tool-setting process and (**b**) flow chart of the tool-setting program.

**Figure 10 micromachines-14-02178-f010:**
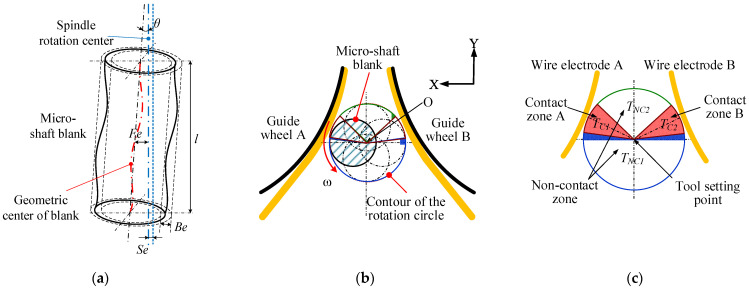
Schematic diagram of the influence of a micro-shaft blank error on the tool-setting process: (**a**) micro-shaft blank error; (**b**) contour of the blank rotation circle; and (**c**) contact area between the micro-shaft blank and wire electrodes.

**Figure 11 micromachines-14-02178-f011:**
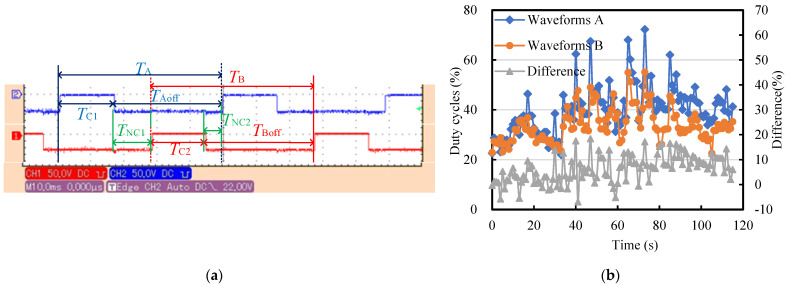
Tool-setting waveform characteristics: (**a**) waveforms collected using an oscilloscope and (**b**) the duty cycle variation and difference of the waveforms.

**Figure 12 micromachines-14-02178-f012:**
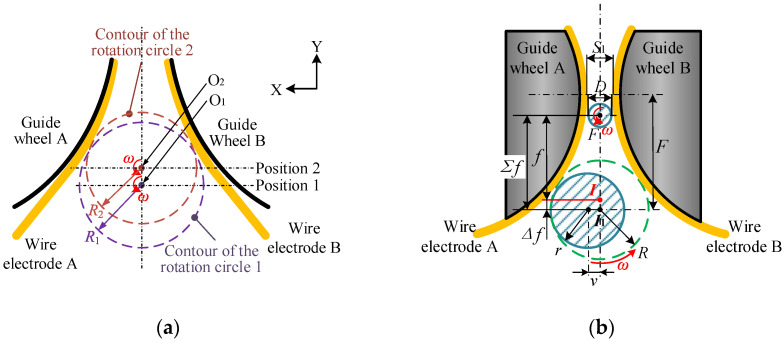
Schematic diagram of the influence of changes in rotation radius on the tool-setting position and the tangential feed distance: (**a**) tool-setting position and (**b**) tangential feed distance.

**Figure 13 micromachines-14-02178-f013:**
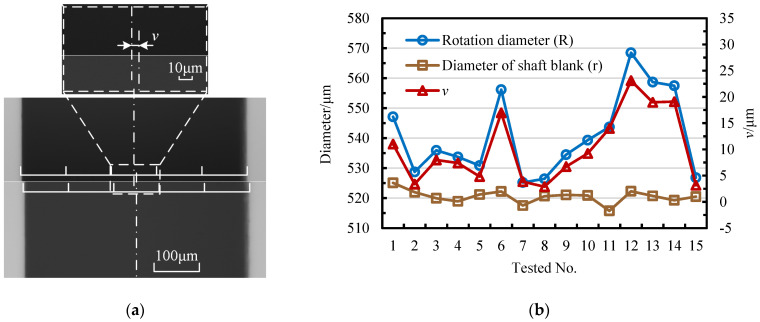
Experimental results of the rotation diameter error of blanks: (**a**) the deviation of the geometric center of blanks and (**b**) the actual diameter, rotation diameter, and deviation of blanks.

**Figure 14 micromachines-14-02178-f014:**
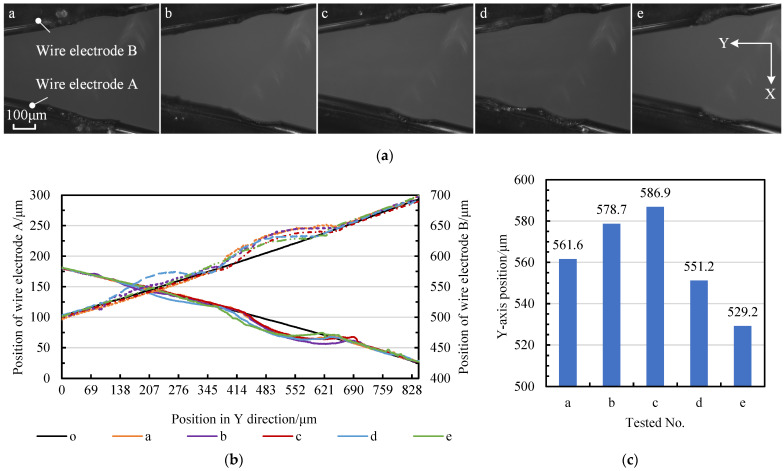
Deviation of tool-setting positions for different micro-shaft blanks: (**a**) CCD photos of the tool-setting positions: a for position one, b for position two, c for position three, d for position four, e for position five; (**b**) the contour curves of the wire electrodes in the tool-setting position; and (**c**) the Y-axis position of the rotation center at different tool-setting positions.

**Figure 15 micromachines-14-02178-f015:**
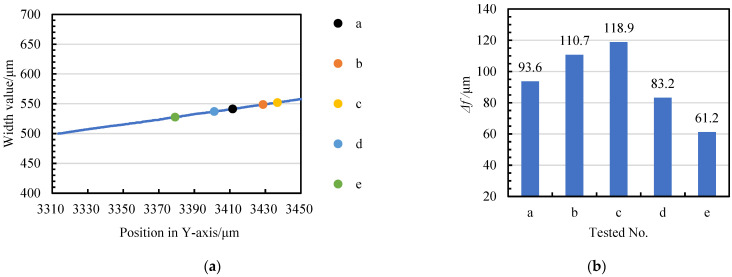
The deviation of tangential feed distance: (**a**) different initial machining position coordinates (a–e corresponds to the five positions in Figure 14 and blue line is the width of narrow slit) and (**b**) compensation value for tangential feed distance Δ*f*.

**Figure 16 micromachines-14-02178-f016:**
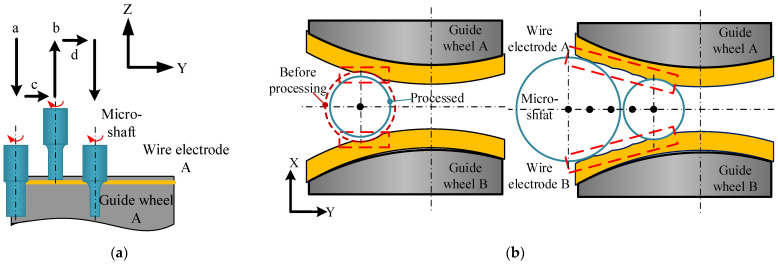
The schematic diagram of wire-electrode wear with the tangential feed of a micro-shaft: (**a**) feed method of micro-shaft machining (The a and b are Z-direction feed steps; the c and d are tangential feed steps) and (**b**) wire-electrode wear.

**Figure 17 micromachines-14-02178-f017:**
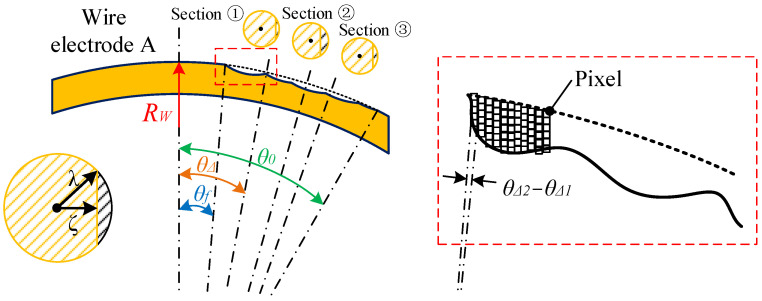
Schematic diagram of wire-electrode wear.

**Figure 18 micromachines-14-02178-f018:**
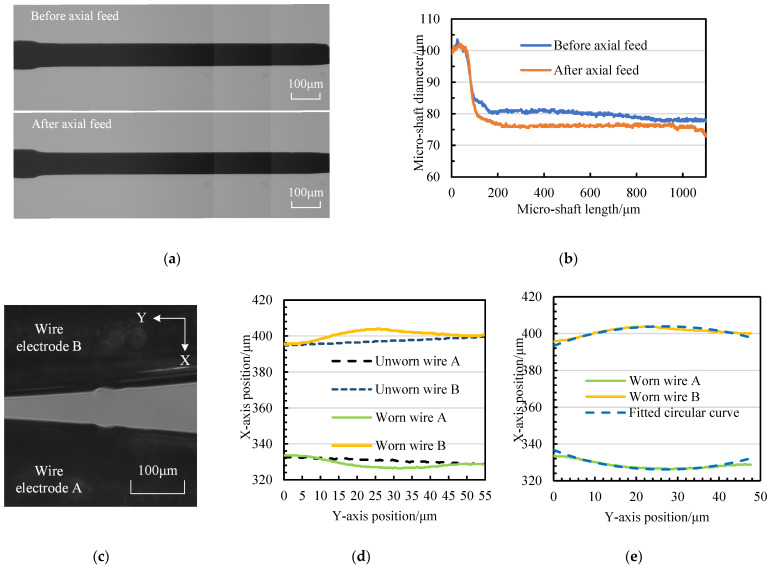
The MRA and WEW of single-axial-fed micro-shafts with diameters of 80 μm in finishing: (**a**) CCD photos of a micro-shaft; (**b**) diameter curves of a micro-shaft; (**c**) CCD photo of a worn-wire electrode; (**d**) boundary curves of a wire electrode; and (**e**) mathematical fitting of WEW.

**Figure 19 micromachines-14-02178-f019:**
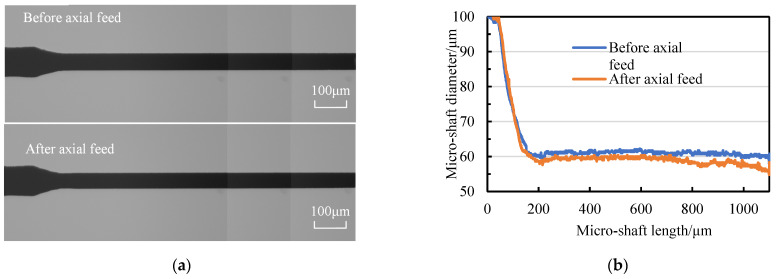
The MRA and WEW of single-axial-fed micro-shafts with diameters of 60 μm in finishing: (**a**) CCD photos of a micro-shaft; (**b**) diameter curves of a micro-shaft; (**c**) CCD photo of a worn-wire electrode; (**d**) boundary curves of a wire electrode; and (**e**) mathematical fitting of WEW.

**Figure 20 micromachines-14-02178-f020:**
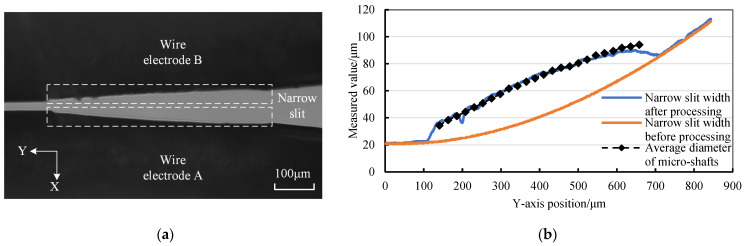
The MRA and WEW for a continuous tangential feed micro-shaft in finishing: (**a**) CCD photo of the worn twin-wire electrode and (**b**) the variation in narrow-slit width and micro-shaft diameter during finishing.

**Figure 21 micromachines-14-02178-f021:**
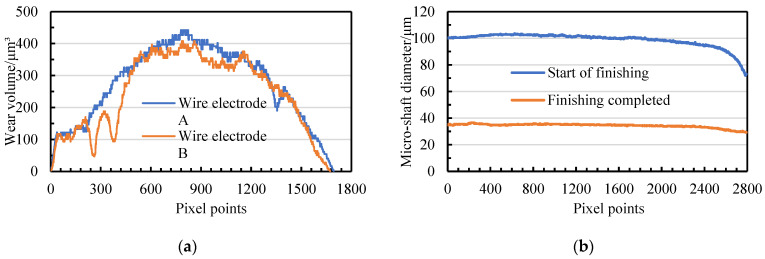
The MRA and WEW after finishing: (**a**) the wear amount of wire electrodes and (**b**) the micro-shaft diameter change.

**Figure 22 micromachines-14-02178-f022:**
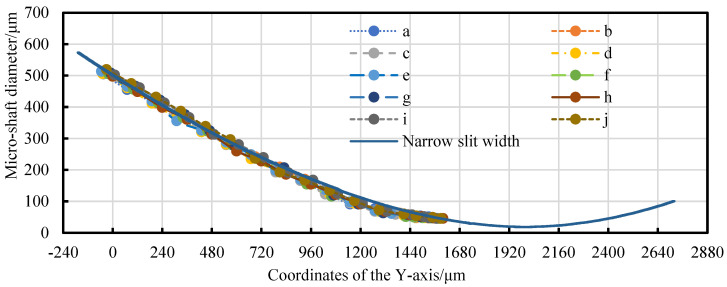
Trends in diameter variation in repetitive machined micro-shafts.

**Figure 23 micromachines-14-02178-f023:**
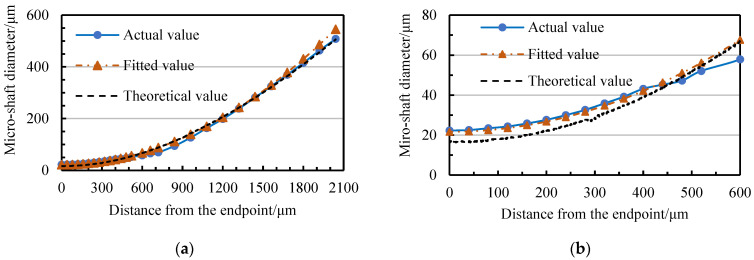
The variation in the theoretical, actual, and fitted value of the micro-shaft diameter: (**a**) 0 < *f* < 2100 μm and (**b**) 1500 < *f* < 2100 μm.

**Figure 24 micromachines-14-02178-f024:**
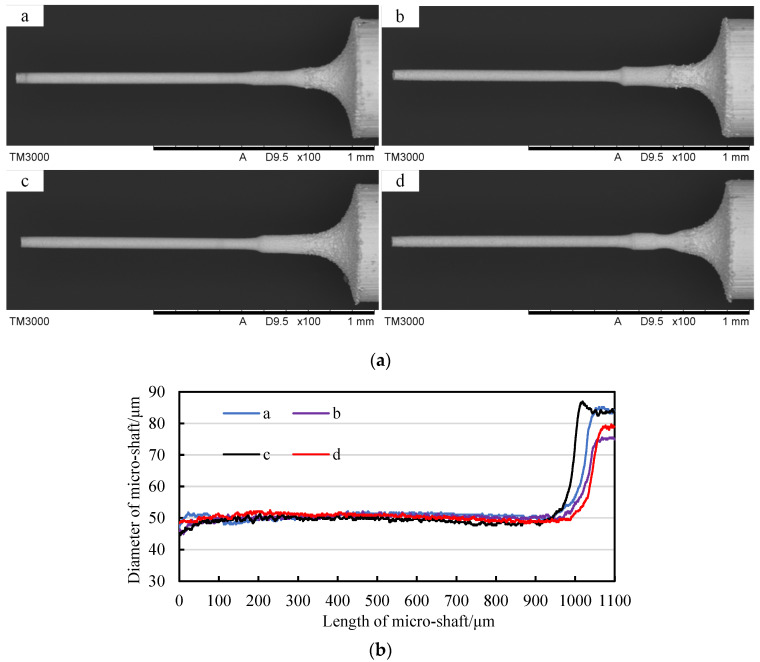
Repetitive machining of micro-shafts with diameters of 50 μm: (**a**) SEM photos of micro-shafts (a for micro-shaft one, b for micro-shaft two, c for micro-shaft three, d for micro-shaft four) and (**b**) the diameter of micro-shafts.

**Figure 25 micromachines-14-02178-f025:**
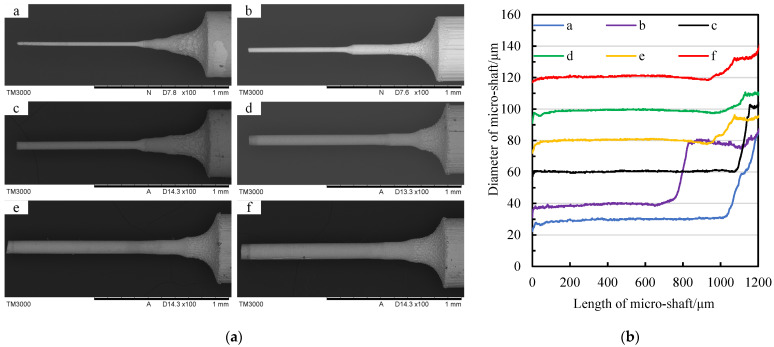
Repetitive machining of micro-shafts with different diameters: (**a**) SEM photos of micro-shafts and (**b**) the diameter of micro-shafts (a for micro-shaft one, b for micro-shaft two, c for micro-shaft three, d for micro-shaft four, e for micro-shaft five, f for micro-shaft six).

**Table 1 micromachines-14-02178-t001:** Processing parameters and conditions of the TMTF-WEDG process.

Parameters		Values		
		Roughing	Semi–Finishing	Finishing
RC pulse power supply	Voltage	120 V	100 V	40 V
	Capacitance	68 nF	470 pF	47 pF
	Resistance	1000 Ω		
Feed rate in Z direction		50 μm/s	30 μm/s	20 μm/s
Feed length per step in the Y direction		30 μm	20 μm	10 μm
Spindle speed		1200 rpm		
Wire speed		Nearly 0.11 mm/s		0
Wire tension		14~15 N		
Dielectric fluid		Deionized water		
Micro-shafts blank		Tungsten (Ø 0.5 mm)		
Wire electrodes		Brass (Ø 0.25^0^_−0.002_ mm) (5 kg/roll) ^1^		

^1^ The deviation within the length range of a roll of brass wire electrode is (−0.002~0) mm.

**Table 2 micromachines-14-02178-t002:** The influence of different narrow-slit widths.

Index Values	Narrow-Slit Width A	Narrow-Slit Width B
Narrow-slit width	*S*_1_ = 18 μm	*S*_1′_ = 40 μm
Tangential feed distance	*f*_1_ ≈ 1600 μm	*f*_2_ ≈ 1800 μm
Initial position difference	Δ_1_ ≈ 50 μm	
Finish position difference	Δ_2_ ≈ 250 μm	

**Table 3 micromachines-14-02178-t003:** Average wire electrode fluctuations with different running linear velocities.

Motor Control Variable Settings	The Linear Velocity of the Wire Electrodes	Average of Fluctuations	
		Wire Electrode A	Wire Electrode B
10	107 μm/s	0.93 μm	2.34 μm
20	213.5 μm/s	2.07 μm	5.65 μm
40	427 μm/s	1.12 μm	3.91 μm
60	640.5 μm/s	0.87 μm	3.47 μm
80	854 μm/s	2.07 μm	6.49 μm
100	1067.5 μm/s	2.45 μm	5.99 μm

**Table 4 micromachines-14-02178-t004:** Tangential feed distance of different micro-shafts.

Micro-Shaft	a	b	c	d
*f*	1517.5 μm			
Δ*f*	125 μm	98 μm	60 μm	116 μm
*Σ* *f*	1642.5 μm	1615.5 μm	1577.5 μm	1633.5 μm

**Table 5 micromachines-14-02178-t005:** Tangential feed distance of different micro-shafts for different target diameters.

Micro-Shaft	a	b	c	d	e	f
Target value	Ø 30 μm	Ø 40 μm	Ø 60 μm	Ø 80 μm	Ø 100 μm	Ø 120 μm
Average value	Ø 29.66 μm	Ø 39.44 μm	Ø 60.35 μm	Ø 79.98 μm	Ø 99.81 μm	Ø 120.29 μm
*f*	1732.9 μm	1611.2 μm	1440.6 μm	1312.4 μm	1204.8 μm	1110.2 μm
Δ*f*	8 μm	162 μm	141 μm	33 μm	65 μm	71 μm
*Σ* *f*	1740.9 μm	1773.2 μm	1581.6 μm	1345.4 μm	1269.8 μm	1181.2 μm

## Data Availability

The data supporting the reported results have been completely declared in this article.

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
