# Peer review of "Research on Geometric Constraint Strategies for Controlling the Diameter of Micro-Shafts Manufactured via Wire Electric Discharge Grinding"

_micromachines, 2023, doi:10.3390/mi14122178_

Round 1

Reviewer 1 Report

Comments and Suggestions for Authors

Conditions Selection: It would be beneficial for the readers, the author could provide more detailed information about the selection of specific conditions in the TMTF-WEDG process. In reference to the spindle speed of 1200rpm, could you briefly justify why this specific speed was chosen for the experiment? Additionally, what criteria were considered when determining the parameters listed in Table 1?

Applications: While you have effectively demonstrated the effectiveness of the geometric constraint strategy in a controlled experimental setting, it would be valuable to discuss potential real-world applications. How might this approach be applied in industries or sectors where micro-shafts are critical components? Providing practical insights could significantly strengthen the paper.

In Figure 11, while it's evident that TNC1 > TNC2, could you provide a brief discussion on the implications and significance of this observation within the context of the theoretical tool-setting waveform analysis?

TA (TC1+TNC1) = TB (TC2+TNC2)≈ 50ms, the unit(ms) should be ms?

It's mentioned that the geometric constraint strategy addresses issues related to the fluctuation of the tool setting position and wire wear. Could you discuss if there are any potential limitations or challenges associated with implementing this strategy that should be taken into consideration?

The results in Figure 25 indeed showcase the ability of the TMTF-WEDG method, supported by the geometric constraint strategy, to consistently achieve micro-shafts with different target diameters. Could you elaborate on the specific implications of this precision control in the context of practical applications or industries where micro-shafts are utilized?

The sample B in Figure 25 is mentioned as a stepped micro-shaft with diameters of 40 μm and 80 μm. Could you discuss if there are any unique considerations or challenges associated with machining stepped micro-shafts compared to those with uniform diameters?

In Table 5, there's detailed information about the average diameter after processing, tangential feed distance, and compensation values for different micro-shafts. Could you provide some insights into how these parameters were determined or optimized for achieving such high accuracy in micro-shaft diameter control?

Are there any specific industries or applications where this level of consistency in micro-shaft diameters is particularly crucial or beneficial?

Limitations and Challenges: It would be beneficial for the readers to know if you encountered any specific difficulties during the implementation of the geometric constraint strategy. Addressing these challenges and discussing potential mitigations would add depth to your research.

Long-Term Durability: Given the importance of micro-shafts in various applications, considering their long-term durability is crucial. Have you conducted any preliminary assessments or simulations regarding the longevity of the micro-shafts produced using this method? Insights into the expected lifespan of these components would be valuable.

Author Response

  • Conditions Selection: It would be beneficial for the readers, the author could provide more detailed information about the selection of specific conditions in the TMTF-WEDG process. In reference to the spindle speed of 1200rpm, could you briefly justify why this specific speed was chosen for the experiment? Additionally, what criteria were considered when determining the parameters listed in Table 1?

The specific conditions in the TMTF-WEDG process, including the parameters listed in Table 1 and the spindle speed, have been studied and reported before.

[1] JianYu Jia, YanQing Wang, ShengQiang Yang, WenHui Li. Study of the Effect of Important Factors on the Diameter Accuracy of Micro-shafts Fabricated by TFTF-WEDG. International Journal of Advanced Manufacturing Technology, 2020, 108, 9-10: 3001-3020.

[2] Yanqing Wang, Mattia Bellotti, Jianyu Jia, Zan Li, Shengqiang Yang, Jun Qian, Dominiek Reynaerts. High-efficiency precision machining of micro rods by twin-mirroring-wire tangential feed electrical discharge grinding. Precision Engineering, 2020, 66:482-495.

The detailed reports can refer to the following link: https://doi.org/10.1007/s00170-020-05412-9 and https://doi.org/10.1016/j.precisioneng.2020.08.015.

  • Applications: While you have effectively demonstrated the effectiveness of the geometric constraint strategy in a controlled experimental setting, it would be valuable to discuss potential real-world applications. How might this approach be applied in industries or sectors where micro-shafts are critical components? Providing practical insights could significantly strengthen the paper.

Thank you for your highly constructive suggestions. The growing demand for micro-parts or parts with micro-structures has promoted the development of micro-fabrication. As an important product and tool in the field of micro-fabrication, the accuracy of the micro-shaft directly affects the operability of the product, especially as a tool for micro-fabrication, such as the processing of micro-holes, micro-grooves, and micro three-dimensional structures on the surface. The accuracy of the structures or parts to be processed depends on the uniformity of the micro-shaft, including the uniformity of the diameter in the axial direction of a single micro-shaft and the repeatability of the diameter for repeatedly processed micro-shafts. These insights have been added to the introduction.

  • In Figure 11, while it's evident that TNC1 > TNC2, could you provide a brief discussion on the implications and significance of this observation within the context of the theoretical tool-setting waveform analysis?

Thank you for your comment. Due to the narrow slit formed by the wire electrode appearing in a trumpet shape in the XY plane, the arc length corresponding to TNC1 is greater than that corresponding to TNC2. Figure 11 intuitively illustrates the influence of shape error and clamping error of the micro-shaft blank on the tool-setting waveform characteristics. The waveform characteristics of the contact between the micro-shaft blanks and the wire electrodes A and B can be used as the standard for tool-setting. When the difference between the average voltage values of two square waves is within 0.5V, the rotation center of the micro-shaft blanks can be considered on the tangential feed path. Thereby, the initial machining position of the micro-shafts is determined, which is crucial for determining the tangential feed distance of the micro-shafts and achieving geometric constraints on the diameter. These discussions have been added to lines 240, 261, and 287, respectively.

  • TA (TC1+TNC1) = TB (TC2+TNC2) ≈ 50ms, the unit(ms) should be ms?

Yes, the unit(ms) should be ms. TA and TB indicate the period of contact between the shaft blank and the wire electrode, which corresponds to the rotational speed of the spindle. The spindle speed is 1200rpm, and the waveform period should theoretically be 50ms.

  • It's mentioned that the geometric constraint strategy addresses issues related to the fluctuation of the tool setting position and wire wear. Could you discuss if there are any potential limitations or challenges associated with implementing this strategy that should be taken into consideration?

Thank you for your question. The ultra-thin disc cutting/grinding tools for microgroove machining on semiconductor wafers can be manufactured by TMTF-WEDG with the geometric constraint strategy. The processed object changes from a micro-shaft to an ultra-thin disc-shaped tool, and the rotation axis also changes from the C-axis (about the Z-axis) to the A-axis (about the X-axis). Because of the significant increase in material removal, the wear of the wire electrodes also increases. Running wire electrodes will be necessary to compensate for wear. To ensure the stability of narrow slit width and machining accuracy, the fluctuation of the wire electrodes in the motion state must be effectively controlled. It can be a potential challenge in extending the application of geometric constraint strategies in TMTF-WEDG.

  • The results in Figure 25 indeed showcase the ability of the TMTF-WEDG method, supported by the geometric constraint strategy, to consistently achieve micro-shafts with different target diameters. Could you elaborate on the specific implications of this precision control in the context of practical applications or industries where micro-shafts are utilized?

Taking array micro-hole processing by micro EDM as an example, the micro-shafts used as a tool electrode can be worn during the discharge process. Therefore, the on-machine high-precision, efficient, and automated repetitive manufacturing of micro-shafts is extremely crucial to dimension compensation. Processing micro-shafts with different target diameters satisfy the needs of micro-hole processing with different diameters.

  • The sample B in Figure 25 is mentioned as a stepped micro-shaft with diameters of 40 μm and 80 μm. Could you discuss if there are any unique considerations or challenges associated with machining stepped micro-shafts compared to those with uniform diameters?

For a stepped micro-shaft manufactured by TMTF-WEDG, the only machining parameter that needs to be changed is the Z-axis feed distance. With the tangential feed of finishing in geometric constraint strategy, the distance of the Z-direction feed is 1200 μm. The distance of the Z-direction feed changes to 800μm when the diameter of the micro-shafts is processed to 80μm. With further tangential feed, constrain the diameter of the micro-shafts to 40 μm within the length range of 800 μm.

  • In Table 5, there's detailed information about the average diameter after processing, tangential feed distance, and compensation values for different micro-shafts. Could you provide some insights into how these parameters were determined or optimized for achieving such high accuracy in micro-shaft diameter control?

The parameters in Table 5 were determined by the details of geometric constraint strategies.

Firstly, the geometric relationship between the width of the narrow slit formed by the twin-mirroring-wire electrodes and the tangential feed distance was determined. For a micro-shaft with a certain target diameter, the tangential feed distance f is determined. Taking the tangential position with a narrow slit width of 500μm as a reference, the deviation Δf between the initial machining position obtained from the tool setting strategy and the reference point can be received. Therefore, for different micro-shaft blanks, the tangential feed distance is equal to the sum of f and Δf. The single-step motion length of a linear motion mechanism driven by a servo motor is 0.1μm, and the accuracy of linear motion is also guaranteed by high-precision guide rails and screws.

  • Are there any specific industries or applications where this level of consistency in micro-shaft diameters is particularly crucial or beneficial?

A typical example is the application of micro-shafts as electrodes for electrical discharge machining of array micro-holes. A certain model of inkjet printer has hundreds of micro spray holes on the orifice plate, and the diameter deviation of micro-holes is required to be controlled within 2μm. The necessary condition to ensure the accuracy of micro-holes is to flexibly and efficiently control the diameter accuracy of the micro-shafts.

  • Limitations and Challenges: It would be beneficial for the readers to know if you encountered any specific difficulties during the implementation of the geometric constraint strategy. Addressing these challenges and discussing potential mitigations would add depth to your research.

Thank you for your comment. The wire electrode fluctuation is a specific difficulty during the geometric constraint strategy by the TMTF-WEDG process for manufacturing ultra-thin disc-cutting tools. Due to the significant increase in material removal, the wear of the wire electrodes also increases, and the strategy of keeping the wire electrode stationary in finishing will no longer be applicable. Running wire electrodes will be necessary to compensate for wear. To ensure machining accuracy, the fluctuation of the line electrode in the motion state must be effectively controlled. The constant tension control system of wire electrodes will be an effective means and a meaningful research direction. The discussion on this viewpoint has been supplemented in lines 486 to 491 of the manuscript.

  • Long-Term Durability: Given the importance of micro-shafts in various applications, considering their long-term durability is crucial. Have you conducted any preliminary assessments or simulations regarding the longevity of the micro-shafts produced using this method? Insights into the expected lifespan of these components would be valuable.

Your guidance is very instructive. At present, we have not conducted a preliminary evaluation or simulation of the lifespan of the micro-shafts produced by this method. We will conduct relevant research in the future, such as the impact of micro features like the electro-etched surface obtained by this processing method on the lifespan when using the micro-shaft as a punch.

Reviewer 2 Report

Comments and Suggestions for Authors

The paper provides a systematic methodology for micro-manufacturing. Minor revisions are needed to further improve the paper's idea and readability.  

Abbreviations TMTF should be defined the first time.

Language errors have been found throughout the paper. Sometimes, the sentence structures are not correct.

Line 18-20. The sentence is too long and should be rewritten properly. For example, “To enable efficient and high-precision machining of micro-shafts with target diameters, without the need for multiple repeated online measurements and reprocessing, a geometric constraint strategy is proposed based on the previously introduced TMTF-WEDG method. This strategy encompasses the tool setting method, tangential feed distance compensation, and an equation that establishes the relationship between tangential distance and diameter variation. These components are derived from a key points analysis of the geometric constraints.”

It is mentioned that the TMTF is used for WEDG which was introduced in the previous research. However, it is difficult to figure out the uniqueness of the current research. The novelty of the research needs to be clearly defined.

The authors claimed that the dimensional deviation in the diameter is 1.5 um while Figure 4 shows a different behavior. Explain.

The relationship between wire fluctuation and shaft diameter deviation needs to be investigated.

Comments on the Quality of English Language

Language errors have been found throughout the paper. Sometimes, the sentence structures are not correct.

Author Response

  • Abbreviations TMTF should be defined the first time.

Please forgive our negligence. The definition of TMTF-WEDG has been clarified in the abstract (line 19) where the abbreviation first appears.

  • Language errors have been found throughout the paper. Sometimes, the sentence structures are not correct.

Sorry for the language errors in the manuscript. The manuscript has been submitted to the MDPI editing service for correction.

  • Line 18-20. The sentence is too long and should be rewritten properly. For example, “To enable efficient and high-precision machining of micro-shafts with target diameters, without the need for multiple repeated online measurements and reprocessing, a geometric constraint strategy is proposed based on the previously introduced TMTF-WEDG method. This strategy encompasses the tool setting method, tangential feed distance compensation, and an equation that establishes the relationship between tangential distance and diameter variation. These components are derived from a key points analysis of the geometric constraints.”

With your more reasonable suggestion, modifications were made to the long sentences from lines 16 to 22.

  • It is mentioned that the TMTF is used for WEDG which was introduced in the previous research. However, it is difficult to figure out the uniqueness of the current research. The novelty of the research needs to be clearly defined.

Thank you for the suggestion. The TMTF-WEDG was indeed proposed in our previous research, and the relevant research reports are as follows:

[1] Jianyu Jia, Yanqing Wang, Zan Li, Shengqiang Yang, Jun Qian, Dominiek Reynaerts. Accuracy improvement of micro-shafts fabricated by the twin-mirroring-wire tangential feed electrical discharge grinding. Journal of Materials Processing Technology, 2020, 286: 116808.

[2] Jianyu Jia, Yanqing Wang, Shengqiang Yang, Wenhui Li. Study on taper reduction of high aspect ratio micro-shafts fabricated by twin-mirroring-wire tangential feed electrical discharge grinding (TMTF-WEDG). Journal of Manufacturing Processes, 2020, 57: 614-629.

[3] JianYu Jia, YanQing Wang, ShengQiang Yang, WenHui Li. Study of the Effect of Important Factors on the Diameter Accuracy of Micro-shafts Fabricated by TFTF-WEDG. International Journal of Advanced Manufacturing Technology, 2020, 108, 9-10: 3001-3020.

[4] Yanqing Wang, Mattia Bellotti, Jianyu Jia, Zan Li, Shengqiang Yang, Jun Qian, Dominiek Reynaerts. High-efficiency precision machining of micro rods by twin-mirroring-wire tangential feed electrical discharge grinding. Precision Engineering, 2020, 66:482-495. 

The manuscript mainly reports the geometric constraint strategy for micro-shafts diameter machined by TMTF-WEDG, which is a recent research result. The novelty of the research has been illustrated as “To avoid multiple repeated online measurements and reprocessing, and facilitate efficient and high-precision machining of target diameter micro-shafts arbitrarily, further research has been carried out based on the TMTF-WEDG method to achieve geometric constraints on the diameter of the micro-shafts and solve several specific problems that affect diameter prediction and control in this process.” in lines 103 to 107.

  • The authors claimed that the dimensional deviation in the diameter is 1.5 um while Figure 4 shows a different behavior. Explain.

In Figure 4, the purpose of experimental data is to clarify a reference for determining the tangential feed distance or final machining position of each stage. The data in Figure 4 is used to test whether the selected processing parameters are applicable. As the experiment (in Figure 4) is a preliminary exploration, the precision control strategies for the diameter of micro-shafts have not yet been adopted. The necessary explanations have been added to lines 145 to 149 of the manuscript.

  • The relationship between wire fluctuation and shaft diameter deviation needs to be investigated.

    Thank you for the guidance. The relationship between wire fluctuation and micro-shaft diameter deviation has been reported in the article “Accuracy improvement of micro-shafts fabricated by the twin-mirroring-wire tangential feed electrical discharge grinding” published in the Journal of Materials Processing Technology. The article can be accessed through the following link: https://doi.org/10.1016/j.jmatprotec.2020.116808 .